# Dual flows with Contrastive Guidance for Generating Highly Designable Proteins

## Abstract

Deep generative models have achieved substantial success in protein design. A prevalent approach for de novo protein design involves initially designing a protein backbone structure using deep generative models, such as diffusion and flow models, followed by using a separate inverse folding model to design the corresponding sequence. Recently, co-design methods, which aim to jointly generate the structure and sequence of a protein, have attracted considerable attention. Despite of this, co-designing sequences and structures of long proteins remains challenging. The complexity of this high-dimensional multimodal generative modeling makes sampling of diffusion and flow models prone to accumulated errors, often leading to non-designable regions. To tackle this challenge, we introduce a contrastive guided sampling algorithm with dual multimodal flows to sample both sequences and structures of highly designable proteins. The contrastive guidance uses the lower-quality flow to help the higher-quality flow avoid non-designable regions by gently steering it during sampling. Our method achieves designability of 80% for length-400 proteins and 37% for length-500 proteins, significantly outperforming previous approaches.

## 1 Introduction

Proteins are fundamental molecules essential to biology. The ability to design novel proteins (Huang et al., 2016) presents a promising pathway for the development of advanced therapeutics (Silva et al., 2019), biomaterials (King et al., 2012), biocatalysis (Röthlisberger et al., 2008), among other applications. Protein engineering has traditionally relied on significant expertise and intensive experimental efforts. This impedes the advancement of novel biotechnologies. Computational approaches incorporating deep learning have transformed the paradigm, markedly accelerating the process of *de novo* protein design (Ding et al., 2022).

Noticeably, deep generative models, such as diffusion (Ho et al., 2020; Song et al., 2021) and flow (Lipman et al., 2022; Liu et al., 2022) models, have been extensively utilized in *de novo* protein design, yielding promising results. Following the fundamental biological principle that structure determines function, numerous efforts have been focused on generating protein backbone structures, i.e., **protein backbone design**. Various protein presentations have been explored in this field, including $C_\alpha$ only (Trippe et al., 2022), backbone torsion angles (Wu et al., 2024), and residue frame representation (Yim et al., 2023b; Lin & Alquraishi, 2023) which is adopted from AlphaFold2 (Jumper et al., 2021) proposed for protein structure prediction. Among these, residue frame representation has demonstrated the best performance (Watson et al., 2023) and has been adopted in recent studies of protein backbone design. Due to the lengthy reverse generative process of diffusion models causing slow inference speed, researchers have shifted to flow models for faster and higher-quality protein backbone generation (Yim et al., 2023a; Bose et al., 2023). Given these well-developed protein backbone design models along with inverse folding models (Dauparas et al., 2022; Hsu et al., 2022), *de novo* protein design can be achieved by initially constructing a protein backbone structure and subsequently designing sequences based on this structure. The inverse folding models (Dauparas et al., 2022) and folding models (Jumper et al., 2021; Lin et al., 2022) can be utilized together as tools for *in silico* evaluation of protein backbone design by comparing the generated structures and the refolded ones. Another line of research focuses on **protein sequence generation**, which models the distribution of protein sequences while ignoring structures. To bypass the complexities of protein design pipelines with multiple models and more effectively capture the intricate relationship

between protein sequences and structures, some researchers have introduced the concept of **protein sequence-structure co-design** (Shi et al., 2022; Campbell et al., 2024), which aims to jointly generate continuous protein structures alongside corresponding discrete amino acid sequences. Representative works include: Protein Generator (Lisanza et al., 2023; 2024), a sequence space diffusion model based on RoseTTAfold (Baek et al., 2021) that simultaneously generates protein sequences and structures; ProtPardelle (Chu et al., 2024), an all-atom Euclidean diffusion model paired with an iterative sequence prediction mechanism; Multiflow (Campbell et al., 2024), a multimodal flow model; ESM3 (Hayes et al., 2024), a frontier multimodal generative language model that reasons over the sequence, structure, and function of proteins. These works introduce a new paradigm and highlight a promising direction for future research.

Despite significant achievements *de novo* protein design based on generative models has achieved, designing long proteins remains challenging. This complexity arises from the vast search space and the inherently challenging nature of high-dimensional generative modeling, as seen in other areas such as image and text generation. Researchers in fields like image and text generation attempt to overcome this challenge by scaling up both the training dataset size and the model parameters (Achiam et al., 2023; Rombach et al., 2022). Acquiring large-scale, high-quality protein structure datasets is challenging, resulting in a lack of effective solutions to the high-dimensional challenges in the field of protein design. The performance of diffusion and flow models are attributed to sampling errors (Xu et al., 2023), which mainly come from two aspects: discretization errors of SDE/ODE samplers and the approximation error of the learned neural network relative to the ground truth drift (e.g., the score function in diffusion models and the vector field in flow models). The errors can be accumulated and further amplified along the sampling process (Li & van der Schaar, 2023). Under the shadow of the aforementioned issues, even the most advanced protein design models are not consistently able to generate high-quality proteins.

We assume that it is not possible to perfectly learn the joint distribution of protein sequence and structure from limited high-quality data due to its intricate, high-dimensional nature and complex structure. By aiming to improve protein sequence-structure co-design from the perspective of sampling rather than training, we propose an approach based on dual flows with contrastive guidance for generating high-designable proteins. This approach calibrates the drift (i.e., vector field) of one flow with that of another during the sampling process. More specifically, we initially pretrain a multimodal flow model following Campbell et al. (2024) and slightly fine-tune it on self-synthesized data and selective self-synthesized data to obtain two flows differing in quality. Note that neither of the dual flows is perfect. Leveraging the contrast of the dual flows, we can calibrate the vector field of higher-quality flow by steering it away from the regions of non-designable samples pointed out by the lower-quality flow. We provide both theoretical (albeit non-formal) justification and intuitive understanding of our work. We implement the contrastive guidance on Euclidean space, SO(3) manifold, and categorical probability space for both the sequences and structures of proteins. With all these together, our method achieves designability of 80% for length-400 proteins and 37% for length-500 proteins, outperforming previous approaches by a significant margin.

## 2 PRELIMINARIES

In this section, we discuss the preliminary information about flow matching for modeling both continuous and discrete variables, which will be used in our method.

In flow matching (Lipman et al., 2022; Liu et al., 2022; Albergo & Vanden-Eijnden, 2022), we consider a predefined *marginal probability path* $p_t$ that interpolates prior distribution $p$ and data distribution $q$, i.e., $p_0 = p$ and $p_1 = q$, to train a generative model that transports a source sample $\mathbf{x}_0 \sim p$ to a target sample $\mathbf{x}_1 \sim q$. We use a *conditional probability path* $p_t(\mathbf{x}|\mathbf{x}_1)$ with $p_0(\cdot|\mathbf{x}_1) \equiv p(\cdot)$ and $p_1(\cdot|\mathbf{x}_1) \approx \delta(\mathbf{x}_1)$ to construct the above $p_t$ by $p_t(\mathbf{x}) = \int p_t(\mathbf{x}|\mathbf{x}_1)q(\mathbf{x}_1)d\mathbf{x}_1$. The above idea of using conditional probability to construct the marginal probability path and train the generative model is utilized in both versions for modeling continuous and discrete variables.

For modeling continuous variables, given a simple implementation of $p_t(\mathbf{x}|\mathbf{x}_1)$ (e.g., Gaussian paths), we can easily derive the *conditional vector field* $u_t(\mathbf{x}|\mathbf{x}_1)$ that satisfies the continuity equation $\dot{p}_t(\mathbf{x}|\mathbf{x}_1) + \mathrm{div}_{\mathbf{x}}[p_t(\mathbf{x}|\mathbf{x}_1)u_t(\mathbf{x}|\mathbf{x}_1)] = 0$, which means $u_t(\cdot|\mathbf{x}_1)$ *generates* $p_t(\cdot|\mathbf{x}_1)$. The

*marginal vector field* can be defined as

$$u_t(\mathbf{x}) = \int u_t(\mathbf{x}|\mathbf{x}_1)\frac{p_t(\mathbf{x}|\mathbf{x}_1)p(\mathbf{x}_1)}{p_t(\mathbf{x})}d\mathbf{x}_1. \tag{1}$$

The key insight of flow matching is that $u_t(\mathbf{x})$ can be proved to generate $p_t(\mathbf{x})$, i.e., satisfying the continuity equation $\dot{p}_t(\mathbf{x}) + \text{div}_\mathbf{x}[p_t(\mathbf{x})u_t(\mathbf{x})] = 0$. Therefore, conditional flow-matching (CFM) loss can be utilized to train a time-dependent neural network $v_\theta$ as follows:

$$\mathcal{L}_{\text{CFM}} = \mathbb{E}_{t,p(\mathbf{x}_1),p_t(\mathbf{x}|\mathbf{x}_1)}\|v_\theta(\mathbf{x}, t) - u_t(\mathbf{x}|\mathbf{x}_1)\|^2. \tag{2}$$

We can generate samples by solving the Ordinary Differential Equation (ODE) as $\frac{d\mathbf{x}_t}{dt} = v_\theta(\mathbf{x}_t, t)$.

For modeling discrete variables, variants of flow matching are proposed, such as Dirichlet Flow Matching (Stark et al., 2024), Discrete Flow Matching (Campbell et al., 2024; Gat et al., 2024) and Fisher Flow Matching (Davis et al., 2024; Cheng et al., 2024). Here we follow the framework of Discrete Flow Matching which is based on Continuous-Time discrete Markov Chain (CTMC). Here $\mathbf{x} \in \{1, \dots, S\}^D$ is a random discrete variable, e.g., a length-$D$ sequence where each element takes on one of $S$ states, and we denote $\boldsymbol{j}$ as a specific state of $\mathbf{x}$.

We first introduce the concept of rate matrix $R_t \in \mathbb{R}^{S \times S}$, which plays a similar role to the marginal vector field in the continuous case. The probability that $\mathbf{x}_t$ jumps to a different state $\boldsymbol{j}$ is $R_t(\mathbf{x}_t, \boldsymbol{j})dt$ after a infintesimal time step $dt$ is $R(\mathbf{x}_t, \boldsymbol{j})dt$, i.e., $p_{t+dt}(\boldsymbol{j}|\mathbf{x}_t) = \delta\{\mathbf{x}_t, \boldsymbol{j}\} + R_t(\mathbf{x}_t, \boldsymbol{j})dt$ where $\delta\{\boldsymbol{i}, \boldsymbol{j}\}$ is an element-wise Kronecker delta which 1 in dimension $d$ when $\boldsymbol{i}^d = \boldsymbol{j}^d$ and 0 otherwise. If a rate matrix $R_t$ and a probability path $p_t$ satisfy the Kolmogorov equation $\partial_t p_t(\mathbf{x}_t) = \sum_{\boldsymbol{j} \neq \mathbf{x}_t} R_t(\boldsymbol{j}, \mathbf{x}_t)p_t(\boldsymbol{j}) - \sum_{\boldsymbol{j} \neq \mathbf{x}_t} R_t(\mathbf{x}_t, \boldsymbol{j})p_t(\mathbf{x}_t)$ (analogous to continuity equation in the continuous case), we say the rate matrix $R_t$ *generates* $p_t(\mathbf{x})$.

In Discrete Flow Matching, we also use the conditional probability path to construct the marginal one as $p_t(\mathbf{x}_t) := \mathbb{E}_{p(x_1)}[p_t(\mathbf{x}_t|\mathbf{x}_1)]$. We define $R_t(\mathbf{x}_t, \boldsymbol{j}|\mathbf{x}_1)$ as a conditional rate matrix that generates $p_t(\mathbf{x}_t|\mathbf{x}_1)$. Notably, $R_t(\mathbf{x}_t, \boldsymbol{j}|\mathbf{x}_1)$ can usually be in a simple formula. For example, it can be that $R_t(\mathbf{x}_t, \boldsymbol{j}|\mathbf{x}_1) := \delta\{\mathbf{x}_1, \boldsymbol{j}\}\delta\{\mathbf{x}_t, M\}/(1 - t)$ where $M$ is the mask token. It can be proved that the marginal rate matrix $R_t(\mathbf{x}_t, \boldsymbol{j}) := \mathbb{E}_{p(\mathbf{x}_1|\mathbf{x}_t)}[R_t(\mathbf{x}_t, \boldsymbol{j}|\mathbf{x}_1)]$ can *generate* the marginal probaility path $p_t(\mathbf{x}_t)$ which we have defined above, where $p_t(\mathbf{x}_1|\mathbf{x}_t) = p(\mathbf{x}_t|\mathbf{x}_1)q(\mathbf{x}_1)/p_t(\mathbf{x}_t)$. Since the posterior $p_t(\mathbf{x}_1|\mathbf{x}_t)$ is intractable, we can use a neural network $p_t^\theta(\mathbf{x}_1|\mathbf{x}_t)$ to approximate it. And we denote $R_t^\theta(\mathbf{x}_t, \cdot) := \mathbb{E}_{p_t^\theta(\mathbf{x}_1|\mathbf{x}_t)}[R_t(\mathbf{x}_t, \cdot|\mathbf{x}_1)]$. Then we can generate a sample by iteratively sampling from the process as

$$p_{t+dt}(\mathbf{x}_{t+dt}|\mathbf{x}_t) = \delta\{\mathbf{x}_{t+dt}, \mathbf{x}_t\} + R_t^\theta(\mathbf{x}_t, \mathbf{x}_{t+dt})dt + o(dt). \tag{3}$$

## 3 METHOD

In this section, we will present our method as illustrated in Fig. 1. In Sec. 3.1, we introduce the pretraining procedure of the multimodal flow model for protein co-design following Campbell et al. (2024). In Sec. 3.2, we offer both theoretical (albeit non-formal) justification and an intuitive understanding of the motivation behind constructing the dual flows. Finally, in Sec. 3.3, we introduce the specific implementation of the proposed contrastive guidance for flow matching on Euclidean space, SO(3) manifold, and categorical probability.

### 3.1 MULTIMODAL FLOW MATCHING FOR PROTEIN DESIGN

We focus on designing both the sequence and structure of proteins. Following Yim et al. (2023b), the protein structure is referred to as the backbone atomic coordinates of each residue and is represented as elements of SE(3) to capture the rigidity of the local frames along the backbone.

We follow the notation in Campbell et al. (2024). Specifically, a protein with $D$ residues is represented as $\mathbf{T} = \{T^d\}_{d=1}^D$, where $T^d = (x^d, r^d, a^d)$ is the residue state of index $d$. Here $x \in \mathbb{R}^3$ is the translation of the residue's $C_\alpha$ atom, $r \in \text{SO}(3)$ is the rotation of the residue's local frame with respect to the global reference frame, and $a \in \{1, \dots, 20\} \cup \{M\}$ is one of 20 amino acids or the mask state $M$. Thus, the goal of protein sequence-structure co-design can be then formalized as a generative task that models the distribution $q(\mathbf{T})$.

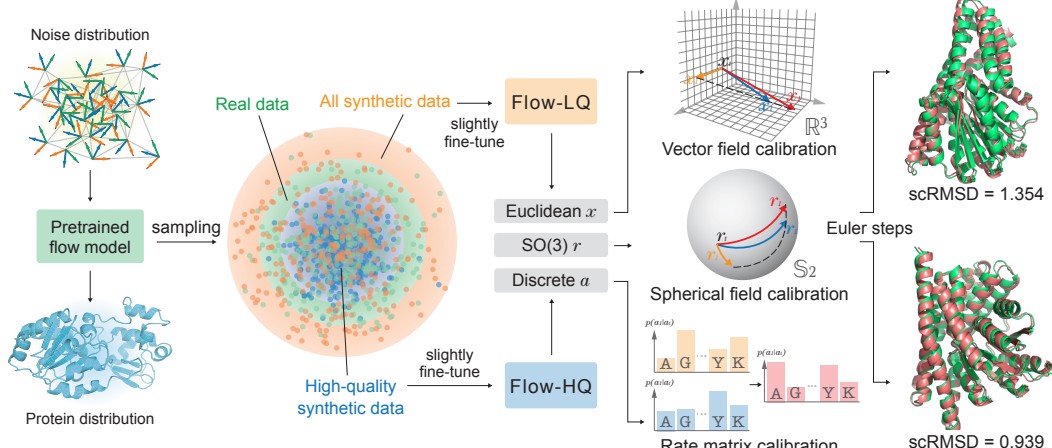

Figure 1: Overview of our method. We initially pretrain a multimodal flow model on real data for protein sequence-structure co-design. We then slightly fine-tune the base flow model on self-synthesized data and selective high-quality self-synthesized data for a few epochs separately to derive two models, named FLOW-LQ and FLOW-HQ. Due to the contrast of the dual flows, the vector fields and rate matrix of FLOW-HQ can be calibrated by being pushed away from the vector fields of FLOW-LQ which are more likely to point towards regions of non-designable samples. The sequence and structure of the protein sampled by our method are highly consistent.

We build our base flow model as introduced by Campbell et al. (2024). We first define the conditional probability path introduced in Sec. 2 as $p_t(\mathbf{T}_t|\mathbf{T}_1) \coloneqq \prod_{d=1}^{D} p_t(x_t^d|x_1^d)p_t(r_t^d|r_1^d)p_t(a_t^d|a_1^d)$.

$p_t(x_t^d|x_1^d)$ and $p_t(r_t^d|r_1^d)$ are implicitly defined by the sampling procedure: $x_t^d = tx_1^d + (1-t)x_0^d$ and $r_t^d = \exp_{r_0^d}(t\log_{r_0^d}(r_1^d))$ where $x_0^d \sim \mathcal{N}(0,I)$ $r_0^d \sim \mathcal{U}_{\text{SO(3)}}$. $\mathcal{U}_{\text{SO(3)}}$ is the uniform distribution over SO(3). exp and log are exponential and logarithm maps. $r_t^d$ can be seen as the linear interpolant along the geodesic path connection $r_0^d$ and $r_1^d$ (Chen & Lipman, 2023). $p_t(a_t^d|a_1^d)$ is defined as $\text{Cat}(t\delta\{a_1^d, a_t^d\} + (1-t)\delta\{M, a_t^d\})$, which means that we linearly interpolate between a probability mass concentrated entirely on the mask token and the data distribution.

With the above conditional probability path, we train a neural network to approximate the vector field on Euclidean space and SO(3) manifold for protein structures and rate matrix for protein sequences. In practice, we use a neural network parameterized with $\theta$ to estimate the vector fields and rate matrix as:

$$\hat{v}_x^d(\mathbf{T}_t) = \frac{\hat{x}_1^d(\mathbf{T}_t) - x_t^d}{1-t}, \ \hat{v}_r^d(\mathbf{T}) = \frac{\log_{r_t^d}(\hat{r}_1^d(\mathbf{T}_t))}{1-t}, \ \hat{R}_t^d(\mathbf{T}_t, j^d) = \frac{\hat{p}(a_1^d = j^d|\mathbf{T}_t)}{1-t}\delta\{a_t^d, M\}, \quad (4)$$

where $\hat{x}_1^d(\mathbf{T}_t)$, $\hat{r}_1^d(\mathbf{T}_t)$, $\hat{p}(a_1^d = j^d|\mathbf{T}_t)$ are translation, rotation, and amino acid type of $d$-th residue at time $t = 1$ predicted by the neural network $\theta$ given the noisy protein $\mathbf{T}_t$ and time $t$ as input.

The overall training objective can be formalized with as follows:

$$\mathcal{L} = \mathbb{E}_{t, \mathbf{T}_1, \mathbf{T}_t}\left[\sum_{d=1}^{D}\left[\frac{\|\hat{x}_1(\mathbf{T}_t) - x_1^d\|^2}{1-t} + \frac{\|\log_{r_t^d}(\hat{r}_1^d(\mathbf{T}_t)) - \log_{r_t^d}(\hat{r}_1^d)\|^2}{1-t} - \log\hat{p}(a_1^d|\mathbf{T}_t)\right]\right], \quad (5)$$

where $t \sim \mathcal{U}(0,1), \mathbf{T}_1 \sim p(\mathbf{T}), \mathbf{T}_t \sim p_t(\mathbf{T}_t|\mathbf{T}_1)$. Note that there are slight adjustments on the coefficients based on time $t$ which we omit related details of derivation here for brevity.

## 3.2 MOTIVATION AND DERIVATION OF DUAL FLOWS

While the base multimodal flow model is powerful in modeling distributions over short proteins, it struggles with generating long proteins effectively. The reasons are multifaceted: approximation errors (i.e., model estimation errors) and discretization errors of the ODE solver (i.e., the sampler), among others (Xu et al., 2023). These errors accumulate throughout the sampling process. Due to

the high-dimensional nature of the problem when generating long proteins, these errors can further deteriorate, potentially causing the sampling trajectories to become lost in the vast probability space and ultimately leading to regions of non-designable samples.

The designable proteins have intrinsic properties and correlation of structures and sequences, which the neural network cannot easily and perfectly learn. Thus, we properly assume that the flow models that correspond to real data distribution $q(\mathbf{T})$ are always unavailable.

We further assume that the synthesized proteins genrated by the pretrained base model as introduced in Sec. 3.1 follows the distribution $\tilde{p}(\mathbf{T}) = p(\mathbf{T}) \exp(-\zeta(\mathbf{T}))$, where $\zeta(\cdot)$ is a function for which we do not know the explicit formula. Note that the above assumption implies certain constraints on $\zeta(\cdot)$ since $\tilde{p}(\mathbf{T})$ is self-normalized, i.e., $\int p(\mathbf{T}) \exp(-\zeta(\mathbf{T})) = 1$, thereby satisfying the property required to be a probability density function.

We curate two small datasets using the self-synthesized proteins generated by the base flow model. Specifically, we use the pretrained base model to generate a small set consisting of both designable and non-designable proteins as the first dataset and then select the designable proteins from the first dataset to construct the second dataset. We slightly fine-tune the pretrained base model on the two datasets separately to derive the dual flow models, denoted as FLOW-LQ (low quality) and FLOW-HQ (high quality). We denote the induced distribution by FLOW-LQ and FLOW-HQ as $p_L(\mathbf{T})$ and $p_H(\mathbf{T})$, respectively. Here we assume $p_L(\mathbf{T}) = p(\mathbf{T}) \exp(-\zeta_L(\mathbf{T}))$ and $p_H(\mathbf{T}) = p(\mathbf{T}) \exp(-\zeta_H(\mathbf{T}))$. Since training generative models on self-synthesized data usually leads to a decrease in quality (Alemohammad et al., 2023), $p_L$ and $p_H$ show worse performance in expectation than native proteins in terms of designability. Nonetheless, due to the the fact that the above two self-synthesized datasets explicitly differ in data quality, we could safely assume that, for any designable protein (denoted as $\mathbf{T}_H$), we have that $p_H(\mathbf{T}_H) > p_L(\mathbf{T}_H)$, and for any non-designable protein (denoted as $\mathbf{T}_L$), we have that $p_H(\mathbf{T}_L) < p_L(\mathbf{T}_L)$. Although we have no knowledge about the explicit formula about $\zeta_L(\cdot)$ and $\zeta_H(\cdot)$, the above claims still indicate that $\zeta_L(\mathbf{T}_H) > \zeta_H(\mathbf{T}_H)$ and $\zeta_L(\mathbf{T}_L) < \zeta_H(\mathbf{T}_L)$.

This inspires us to sample highly designable proteins utilizing the difference between these two flows. Our goal is to sample following the probability defined by

$$\bar{p}(\mathbf{T}) \propto p_H(\mathbf{T}) \left[ \frac{p_H(\mathbf{T})}{p_L(\mathbf{T})} \right]^\lambda \propto p(\mathbf{T}) \frac{\exp\left( - (1 + \lambda)\zeta_H(\mathbf{T}) \right)}{\exp(-\lambda \zeta_L(\mathbf{T}))}, \tag{6}$$

where $\lambda > 0$ is a constant hyperparameter.

It is straightforward to prove that $\bar{p}(\mathbf{T}_H) > p(\mathbf{T}_H)$ and $\bar{p}(\mathbf{T}_L) < p(\mathbf{T}_L)$, which means that the calibrated target distribution $\bar{p}$ puts more (resp. less) probability mass on designable (resp. non-designable) samples than the real-data underlying distribution $p$. Therefore, sampling according to Eq. (6) leads to highly designable proteins, even when all the models we utilize perform worse than the ideal generator corresponds to the ground-truth data distribution, which is hardly available in practice due to the challenges we have discussed at the beginning of this subsection.

## 3.3 CONTRASTIVE GUIDANCE

We will show how to sample according to the distribution in Eq. (6). We start with the flow over Euclidean space. Song et al. (2021) have shown the relationship between the probability ODE flows (whose drift is vector field) and the score-based generative models (whose drift is score function) by Kolmogorov's forward equation (Fokker-Planck equation) (Øksendal, 2003). Zheng et al. (2023) have also show the relationship between vector fields and score function. In our case, similarly, we can also relate the estimated vector field over Euclidean space to the underlying score function

$$v_x^d(\mathbf{T}_t) = \frac{1}{t} x_t^d + \frac{1 - t}{t} \nabla_{x_t^d} \log p_t(\mathbf{T}_t). \tag{7}$$

We define the calibrated marginal probability path at time $t$ as

$$\bar{p}_t(\mathbf{T}_t) \propto p_{H,t}(\mathbf{T}_t)[p_{H,t}(\mathbf{T}_t)/p_{L,t}(\mathbf{T}_t)]^\lambda, \tag{8}$$

where $p_{H,t}$ (resp. $p_{L,t}$) corresponds to the marginal probability path induced by FLOW-HQ (resp. FLOW-LQ) at time $t$.

By applying the score function to both sides in Eq. (8), we have

$$\nabla_{x_t^d} \log \bar{p}_t(\mathbf{T}_t) = (1 + \lambda)\nabla_{x_t^d} \log \bar{p}_{\mathrm{H},t}(\mathbf{T}_t) - \lambda\nabla_{x_t^d} \log \bar{p}_{\mathrm{L},t}(\mathbf{T}_t). \tag{9}$$

We define the calibrated vector field as

$$\bar{v}_x(\mathbf{T}_t) \coloneqq (1 + \lambda)v_x^{\mathrm{H}}(\mathbf{T}_t) - \lambda v_x^{\mathrm{L}}(\mathbf{T}_t), \tag{10}$$

where $v_x^{\mathrm{H}}$ (resp. $v_x^{\mathrm{L}}$) is the vector field of FLOW-HQ (resp. FLOW-LQ), and we ignore residue index $d$ for brevity without ambiguity.

By plugging Eq. (7) (the linearity relationship between the vector field and the score function of the underlying distribution) into both the left and right hand side, we can derive that $\bar{v}_x(\mathbf{T}_t)$ generate the marginal probability $\bar{p}_t(\mathbf{T}_t)$, which justifies that we can use the calibrated vector field to approximately sample from the calibrated target distribution.

Notably, Eq. (10) resembles the formula of classifier-free diffusion guidance (Ho & Salimans, 2022). Likewise, Eq. (10) also holds straightforward meanings. Standing on a data point $\mathbf{T}_t$, the vector field of FLOW-HQ can be calibrated by adjusting its direction away from the vector field of FLOW-LQ, because the latter points more to the regions of non-designable samples. The behind mechanism motivates us to name the method as contrastive guidance.

In practice, we calibrate the estimated "clean" samples (the predicted $\hat{x}_1$) instead of directly on vector fields. Specifically, by plugging Eq. (4) into Eq. (10), we can easily derive that

$$\bar{v}_x(\mathbf{T}_t) = \frac{(1 + \lambda)\hat{x}_1^{\mathrm{H}}(\mathbf{T}_t) - \lambda\hat{x}_1^{\mathrm{L}}(\mathbf{T}_t) - x_t}{1 - t}, \tag{11}$$

where $\hat{x}_1^{\mathrm{H}}(\mathbf{T}_t)$ (resp. $\hat{x}_1^{\mathrm{L}}(\mathbf{T}_t)$) is the frame translation of the predicted "clean" sample given the noisy protein $\mathbf{T}_t$. Eq. (11) implies that calibrating the vector fields of FLOW-HQ and FLOW-LQ as Eq. (10) is equivalent to directly calibrating the predicted "clean" sample as

$$\bar{x}_1(\mathbf{T}_t) \coloneqq (1 + \lambda)\hat{x}_1^{\mathrm{H}}(\mathbf{T}_t) - \lambda\hat{x}_1^{\mathrm{L}}(\mathbf{T}_t) = (1 + \lambda)(\hat{x}_1^{\mathrm{H}}(\mathbf{T}_t) - \hat{x}_1^{\mathrm{L}}(\mathbf{T}_t)) + \hat{x}_1^{\mathrm{L}}(\mathbf{T}_t). \tag{12}$$

We follow the above intuition to calibrate the vector field for the rotation of residue frames. Therefore, we can extend Eq. (12) from Euclidean space to SO(3) manifold and define

$$\bar{r}_1(\mathbf{T}_t) \coloneqq \exp_{\hat{r}_1^{\mathrm{L}}(\mathbf{T}_t)}\left((1 + \lambda)\log_{\hat{r}_1^{\mathrm{L}}(\mathbf{T}_t)}\left(\hat{r}_1^{\mathrm{H}}(\mathbf{T}_t)\right)\right) \quad \text{and} \quad \bar{v}_r(\mathbf{T}) \coloneqq \frac{\log_{r_t}(\bar{r}_1(\mathbf{T}_t))}{1 - t}. \tag{13}$$

where $\hat{r}_1^{\mathrm{H}}(\mathbf{T}_t)$ (resp. $\hat{r}_1^{\mathrm{L}}(\mathbf{T}_t)$) is the frame rotation of the predicted "clean" sample given the noisy protein $\mathbf{T}_t$. The geometric understanding behind Eq. (13) is that FLOW-LQ subtly pushes FLOW-HQ away along the geodesic path connecting their predicted $\hat{r}_1$. We use $\bar{v}_r(\mathbf{T})$ defined in Eq. (13) as the calibrated vector field on SO(3) manifold.

The contrastive guidance for the discrete case (i.e., the sequence of the protein) is slightly different from the continuous case (i.e., the structure of the protein) due to the lack of definition of vector field and score function as in Eq. (7). However, as the vector field and score function can describe how the sample or marginal probability changes as time evolves, we turn to the transition probability with similar meanings in CTMC and define the target transition probability that we want to sample from as follows:

$$\bar{p}(a_{t+dt}|\mathbf{T}_t) \propto p_{\mathrm{H}}(a_{t+dt}|\mathbf{T}_t)[p_{\mathrm{H}}(a_{t+dt}|\mathbf{T}_t)/p_{\mathrm{L}}(a_{t+dt}|\mathbf{T}_t)]^{\lambda}. \tag{14}$$

For brevity and clarity, we rewrite the above equation as follows:

$$\bar{p}(a_{t+dt}{=}i|a_t{=}j) \propto p_{\mathrm{H}}(a_{t+dt}{=}i|a_t{=}j)[p_{\mathrm{H}}(a_{t+dt}{=}i|a_t{=}j)/p_{\mathrm{L}}(a_{t+dt}{=}i|a_t{=}j)]^{\lambda}, \tag{15}$$

where $i, j$ are specific amino acid types. We omit the residue index $d$ and the complete context, i.e., the noisy residues $\mathbf{T}_t$, though we indeed provide them with the neural network as input. Since Eq. (15) resembles the predictor-free guidance proposed by Nisonoff et al. (2024), we can leverage their theoretical results and define the calibrated rate matrix as follows:

$$\bar{R}_t(j, i) = R_{\mathrm{H},t}(j, i)^{1+\lambda} R_{\mathrm{L},t}(j, i)^{-\lambda}, \qquad \text{for } i \neq j, \tag{16}$$

where $R_{\mathrm{H},t}(i,j)$ is transition rate from state $i$ to state $j$ at time $t$ estimated by FLOW-HQ. We have $\bar{R}_t(i,i) = -\sum_{j \neq i} \bar{R}_t(i,j)$ by the definition of the rate matrix.

In our case, since the conditional rate matrix describes a masking process, i.e,

$$R_t(a_t, j|a_1) = \frac{\delta\{a_1, M\}\delta\{a_t, M\}}{1-t}. \tag{17}$$

Thus, we have

$$\hat{R}_t(a_t, j) = \mathbb{E}_{\hat{p}(a_1=j|a_t)}[R_t(a_t, j|a_1=j)] = \frac{\hat{p}(a_1=j|a_t)}{1-t}\delta\{a_t, M\}. \tag{18}$$

It is trivial to verify that the constraint induced by the definition of rate matrix is automatically satisfied, i.e., $\sum_j \hat{R}_t(a_t, j) \equiv 1$ for all $a_t$. So we calibrate the predicted posterior in an equivalent way to Eq. (16) as follows:

$$\bar{p}(a_1=j|a_t) \propto \hat{p}_{\mathrm{H}}(a_1=j|a_t)^{1+\lambda}\hat{p}_{\mathrm{L}}(a_1=j|a_t)^{-\lambda}, \tag{19}$$

which can be efficiently computed because we only consider 20 types of amino acids. Finally, we arrive at the calibrated transition probability $\bar{p}(a_{t+dt}|a_t)$ expressed with

$$\bar{p}(a_{t+dt}|a_t) = \sum_j \bar{p}(a_1=j|a_t)p(a_{t+dt}|a_1=j, a_t). \tag{20}$$

Therefore, at each Euler step, to sample $a_{t+dt}$ from the distribution $\bar{p}(a_{t+dt}|a_t)$, we can sample $a_1, a_{t+dt}$ form the joint distribution $\bar{p}(a_1|a_t)p(a_{t+dt}|a_1, a_t)$, and keep only the $a_{t+dt}$ part of this joint sample. In other words, from time $t$ to $t + dt$, the operation that first unmasks all tokens and then random mask parts of them is equivalent to the operation that randomly selects the same ratio of tokens to unmask. A benefit that we calibrated on the predicted posterior as in Eq. (19) instead of rate matrix or transition probability brings is that we can seamlessly introduce purity sampling scheme (Tang et al., 2022) to the sampling process with contrastive guidance. Specifically, from time $t$ to $t + dt$, we select residues with the top-$k$ values of $\bar{p}(a_1=j|a_t)$ to unmask, where $k = \mathrm{Bin}(n_t, \frac{dt}{1-t})$, $\mathrm{Bin}(\cdot, \cdot)$ is a binomial distribution, and $n_t$ is the number of mask tokens. The purity sampling considers the confidence of the model prediction and has a great positive impact on performance empirically.

## 4 EXPERIMENT

### 4.1 EXPERIMENTAL SETUP

**Dataset.** We strictly follow the settings in Campbell et al. (2024) to train our base flow model. The training data used in Campbell et al. (2024) includes the data used in Yim et al. (2023b) for a total of 18,684 proteins with length 60-384 (which are originally from Protein Data Bank (PDB) (Berman et al., 2000)) and extra 4,179 examples which are distilled from ProteinMPNN (Dauparas et al., 2022). The details about the distillation dataset can be found in Campbell et al. (2024). We use the pretraiend base flow model to generate around 6,000 proteins with length 400 as the fine-tuning dataset for FLOW-LQ. And we select the designable proteins from the 6,000 proteins, resulting in a smaller but higher-quality dataset, which is used for fine-tuning FLOW-HQ. Both FLOW-HQ and FLOW-LQ are fine-tuned for 3 epochs, which takes less than 1 hour on 4 A100 Nvidia GPUs using the AdamW optimizer (Loshchilov, 2017) with learning rate 0.0001. The fine-tuning settings (e.g., optimizer configuration) are inherited from the training settings of Multiflow (Campbell et al., 2024) and are not specifically tuned for out setting.

**Baselines.** To comprehensively evaluate the ability for *de novo* protein design, we compare our method with three types of baselines: backbone design methods, sequence design methods, and co-design methods. The backbone design methods include: **RFdiffusion** (Watson et al., 2023), a model derived from fine-tuning the RoseTTAFold structure prediction network on protein structure denoising tasks; **Chroma** (Ingraham et al., 2023), which introduces a diffusion process that upholds the conformational statistics of polymer ensembles and employs an efficient graph neural network for feature extraction; **FrameDiff** (Yim et al., 2023b) and **Genie** (Lin & Alquraishi,

Table 1: Summary of evaluation metrics on length-400 protein design.

| | scTM ↑ | scRMSD (Å) ↓ | Des.* ↑ | Des. ↑ | ppl ↓ | pLDDT ↑ | Diversity ↓ | Novelty ↓ |
|---|---|---|---|---|---|---|---|---|
| | | | | **length 400** | | | | |
| Native PDBs | 0.84 ± 0.20 | 5.56 ± 7.68 | 0.53 | 0.91 | 7.82 ± 5.03 | 77.72 ± 13.65 | 0.27 ± 0.02 | N/A |
| 1-RFdiffusion | 0.63 ± 0.26 | 10.49 ± 7.62 | 0.22 | 0.64 | N/A | N/A | 0.34 ± 0.02 | 0.79 ± 0.05 |
| 1-FrameFlow | 0.55 ± 0.24 | 14.55 ± 10.67 | 0.10 | 0.53 | N/A | N/A | 0.32 ± 0.02 | 0.79 ± 0.05 |
| 1-Chroma | 0.50 ± 0.16 | 14.44 ± 5.85 | 0.00 | 0.35 | N/A | N/A | 0.37 ± 0.04 | 0.77 ± 0.06 |
| 1-FrameDiff | 0.61 ± 0.24 | 12.82 ± 12.25 | 0.03 | 0.68 | N/A | N/A | 0.49 ± 0.06 | 0.77 ± 0.04 |
| 1-FoldFlow-SFM | 0.34 ± 0.09 | 20.31 ± 7.10 | 0.00 | 0.04 | N/A | N/A | 0.35 ± 0.00 | 0.72 ± 0.01 |
| 1-FoldFlow-Base | 0.31 ± 0.08 | 23.07 ± 8.23 | 0.00 | 0.01 | N/A | N/A | N/A | N/A |
| 1-FoldFlow-OT | 0.34 ± 0.10 | 19.92 ± 6.72 | 0.00 | 0.07 | N/A | N/A | 0.35 ± 0.03 | 0.70 ± 0.08 |
| 1-Genie | 0.21 ± 0.02 | 30.02 ± 4.00 | 0.00 | 0.00 | N/A | N/A | N/A | N/A |
| 8-RFdiffusion | 0.86 ± 0.15 | 3.66 ± 3.78 | 0.48 | 0.96 | N/A | N/A | 0.35 ± 0.02 | 0.80 ± 0.05 |
| 8-FrameFlow | 0.76 ± 0.19 | 6.12 ± 4.94 | 0.30 | 0.89 | N/A | N/A | 0.32 ± 0.02 | 0.77 ± 0.05 |
| 8-Chroma | 0.73 ± 0.15 | 6.51 ± 4.47 | 0.09 | 0.91 | N/A | N/A | 0.32 ± 0.03 | 0.76 ± 0.05 |
| 8-FrameDiff | 0.75 ± 0.17 | 6.09 ± 4.70 | 0.16 | 0.90 | N/A | N/A | 0.48 ± 0.06 | 0.76 ± 0.04 |
| 8-FoldFlow-SFM | 0.42 ± 0.08 | 15.02 ± 3.14 | 0.00 | 0.12 | N/A | N/A | 0.36 ± 0.04 | 0.75 ± 0.02 |
| 8-FoldFlow-Base | 0.39 ± 0.07 | 16.35 ± 3.02 | 0.00 | 0.06 | N/A | N/A | 0.37 ± 0.01 | 0.74 ± 0.03 |
| 8-FoldFlow-OT | 0.43 ± 0.09 | 14.75 ± 3.08 | 0.00 | 0.23 | N/A | N/A | 0.35 ± 0.03 | 0.74 ± 0.06 |
| 8-Genie | 0.24 ± 0.02 | 24.85 ± 1.31 | 0.00 | 0.00 | N/A | N/A | N/A | N/A |
| ProGen2 | N/A | N/A | N/A | N/A | 4.51 ± 3.57 | 57.80 ± 20.92 | N/A | N/A |
| EvoDiff | N/A | N/A | N/A | N/A | 16.78 ± 1.53 | 33.75 ± 11.27 | N/A | N/A |
| DPLM | N/A | N/A | N/A | N/A | 3.50 ± 1.45 | 88.12 ± 8.98 | N/A | N/A |
| ProteinGenerator | 0.78 ± 0.21 | 11.85 ± 7.92 | 0.05 | 0.83 | 7.38 ± 1.31 | 61.52 ± 15.12 | 0.48 ± 0.02 | 0.81 ± 0.04 |
| ProtPardelle | 0.50 ± 0.13 | 34.22 ± 13.21 | 0.00 | 0.42 | 5.57 ± 1.08 | 43.11 ± 10.94 | 0.35 ± 0.06 | 0.80 ± 0.05 |
| Multiflow | 0.93 ± 0.07 | 2.71 ± 3.65 | 0.68 | 0.98 | 7.53 ± 1.77 | 80.18 ± 8.94 | 0.37 ± 0.04 | 0.82 ± 0.04 |
| ESM3 | 0.64 ± 0.27 | 33.01 ± 30.22 | 0.15 | 0.53 | 6.78 ± 2.98 | 69.88 ± 17.94 | 0.28 ± 0.06 | 0.89 ± 0.08 |
| Ours | 0.95 ± 0.05 | 1.99 ± 2.14 | 0.80 | 0.99 | 6.38 ± 1.14 | 80.92 ± 5.87 | 0.44 ± 0.04 | 0.82 ± 0.05 |

2023), which are based on diffusion and have demonstrated commendable generative capabilities; **FrameFlow** (Yim et al., 2023a) and **FoldFlow** (Bose et al., 2023), which engage in protein backbone generation through flow matching. Note that there are three versions of Fold-Flow: **FoldFlow-Base** which utilizes standard flow matching similar to FrameFlow; **FoldFlow-SFM** which extends to a stochastic flow; **FoldFlow-OT** which uses optimal transport data coupling instead of independent data coupling for better training of flow matching. Notably, the inference steps of the above flow models are generally fewer than those of diffusion models.

The sequence design methods include: **Pro-Gen2** (Nijkamp et al., 2023) which is an autoregressive protein language model (we use its 700M version); **EvoDiff** (Alam-dari et al., 2023) which is designed as an order-agnostic autoregressive diffusion model; **DPLM** (Wang et al., 2024) which employs a diffusion language model for sequence generation. The co-design methods include: **ProteinGenerator** (Lisanza et al., 2023; 2024), a sequence space diffusion model based on RoseTTAfold (Baek et al., 2021) that can sample from the joint distribution of protein sequences and structures; **ProtPardelle** (Chu et al., 2024), an all-atom Euclidean diffusion model with an iterative sequence prediction mechanism; **Multiflow** (Campbell et al., 2024), a multimodal flow model; **ESM3** (Hayes et al., 2024), a generative masked language models that model both sequence and tokenized struc-

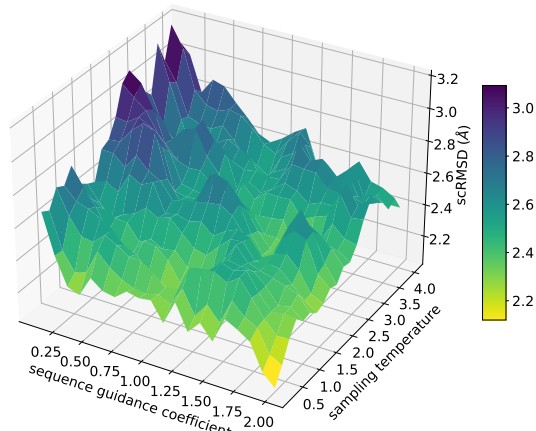

Figure 2: Effects of different sequence guidance coefficients and sampling temperatures on scRMSD (Å).

tures. Additionally, to facilitate comparison with natural proteins (denoted as **Native PDBs** in our tables), we filtered Protein Data Bank (PDB) to construct a standardized comparison dataset. Similar

Table 2: Summary of evaluation metrics on length-500 protein design.

| | length 500 | | | | | | | |
|---|---|---|---|---|---|---|---|---|
| | scTM ↑ | scRMSD (Å) ↓ | Des.* ↑ | Des. ↑ | ppl ↓ | pLDDT ↑ | Diversity ↓ | Novelty ↓ |
| Native PDBs | 0.84 ± 0.19 | 5.47 ± 7.24 | 0.50 | 0.94 | 6.70 ± 4.47 | 77.17 ± 13.93 | 0.26 ± 0.02 | N/A |
| 1-RFdiffusion | 0.57 ± 0.23 | 13.41 ± 7.61 | 0.12 | 0.59 | N/A | N/A | 0.34 ± 0.03 | 0.76 ± 0.05 |
| 1-FrameFlow | 0.47 ± 0.19 | 18.94 ± 12.69 | 0.04 | 0.31 | N/A | N/A | 0.31 ± 0.02 | 0.77 ± 0.06 |
| 1-Chroma | 0.47 ± 0.17 | 16.90 ± 6.57 | 0.01 | 0.28 | N/A | N/A | 0.32 ± 0.01 | 0.75 ± 0.06 |
| 1-FrameDiff | 0.42 ± 0.23 | 25.38 ± 20.28 | 0.00 | 0.36 | N/A | N/A | 0.49 ± 0.05 | 0.75 ± 0.04 |
| 1-FoldFlow-SFM | 0.31 ± 0.07 | 25.78 ± 11.35 | 0.00 | 0.00 | N/A | N/A | N/A | N/A |
| 1-FoldFlow-Base | 0.29 ± 0.07 | 25.90 ± 12.10 | 0.00 | 0.00 | N/A | N/A | N/A | N/A |
| 1-FoldFlow-OT | 0.30 ± 0.07 | 24.39 ± 9.62 | 0.00 | 0.01 | N/A | N/A | N/A | N/A |
| 1-Genie | 0.21 ± 0.02 | 31.44 ± 3.27 | 0.00 | 0.00 | N/A | N/A | N/A | N/A |
| 8-RFdiffusion | 0.76 ± 0.19 | 6.71 ± 5.54 | 0.29 | 0.87 | N/A | N/A | 0.33 ± 0.02 | 0.76 ± 0.04 |
| 8-FrameFlow | 0.66 ± 0.19 | 9.78 ± 5.83 | 0.15 | 0.76 | N/A | N/A | 0.32 ± 0.02 | 0.75 ± 0.05 |
| 8-Chroma | 0.71 ± 0.18 | 8.25 ± 5.73 | 0.01 | 0.81 | N/A | N/A | 0.29 ± 0.01 | 0.76 ± 0.05 |
| 8-FrameDiff | 0.57 ± 0.23 | 15.61 ± 15.53 | 0.01 | 0.66 | N/A | N/A | 0.40 ± 0.06 | 0.74 ± 0.04 |
| 8-FoldFlow-SFM | 0.37 ± 0.05 | 18.39 ± 2.97 | 0.00 | 0.02 | N/A | N/A | N/A | N/A |
| 8-FoldFlow-Base | 0.37 ± 0.05 | 18.37 ± 3.24 | 0.00 | 0.02 | N/A | N/A | N/A | N/A |
| 8-FoldFlow-OT | 0.37 ± 0.06 | 17.66 ± 2.70 | 0.00 | 0.02 | N/A | N/A | N/A | N/A |
| 8-Genie | 0.24 ± 0.01 | 26.84 ± 1.61 | 0.00 | 0.00 | N/A | N/A | N/A | N/A |
| ProGen2 | N/A | N/A | N/A | N/A | 4.27 ± 3.60 | 54.30 ± 18.79 | N/A | N/A |
| EvoDiff | N/A | N/A | N/A | N/A | 16.51 ± 3.82 | 32.94 ± 9.76 | N/A | N/A |
| DPLM | N/A | N/A | N/A | N/A | 3.33 ± 1.80 | 82.57 ± 12.53 | N/A | N/A |
| ProteinGenerator | 0.41 ± 0.19 | 33.91 ± 15.10 | 0.00 | 0.19 | 7.07 ± 1.96 | 44.22 ± 8.72 | 0.45 ± 0.03 | 0.80 ± 0.04 |
| ProtPardelle | 0.41 ± 0.10 | 41.24 ± 10.85 | 0.00 | 0.23 | 4.83 ± 0.80 | 36.62 ± 7.34 | 0.34 ± 0.03 | 0.76 ± 0.05 |
| Multiflow | 0.83 ± 0.15 | 8.48 ± 7.02 | 0.24 | 0.92 | 6.95 ± 1.46 | 69.73 ± 10.61 | 0.35 ± 0.02 | 0.79 ± 0.04 |
| ESM3 | 0.57 ± 0.24 | 37.74 ± 25.22 | 0.03 | 0.49 | 6.90 ± 3.40 | 62.62 ± 15.89 | 0.21 ± 0.03 | 0.86 ± 0.11 |
| Ours | 0.86 ± 0.11 | 4.98 ± 3.58 | 0.37 | 0.99 | 6.01 ± 0.94 | 71.70 ± 8.90 | 0.40 ± 0.03 | 0.77 ± 0.05 |

to FrameDiff (Yim et al., 2023b), we initially filtered out structural data with resolutions $< 5$Å. Subsequently, we employed DSSP (Kabsch & Sander, 1983) to eliminate structures with loop regions exceeding $50\%$. Following this, we categorized the structures based on their lengths. In particular, for proteins with lengths ranging from 450 to 550 residues, we categorized them into the 500-length category for statistical purposes. Similarly, for proteins ranging from 350 to 450 residues, we grouped them into the 400-length category. Subsequently, for each set of proteins within every length category, we de-duplicated using TMalign (Zhang & Skolnick, 2005) and randomly selected 50 proteins for evaluation.

**Evaluation.** We evaluate the ability of *de novo* long protein design of all the baselines and our method from both perspectives of sequences and structures. In order to evaluate the designability of the generated proteins, for backbone design methods, we used ProteinMPNN (Dauparas et al., 2022) and ESMFold (Lin et al., 2022) to refold the generated protein structures, and evaluated the model's designability through two self-consistency metrics (**scTM** and **scRMSD**) over 100 generated proteins each on lengths {400, 500}.

We use two versions of evaluation protocols for backbone design methods: refold once (with "**1-**" as the prefix of method name) and refold 8 times and select the best scRMSD/scTM to report and calculate the related metrics (with "**8-**" as the prefix of method name). We consider two standards for considering a generated protein is designable: scTM>0.5 and more stringent scRMSD<2.0 Å. We report the ratio of designable proteins over all the generated samples as **Des.*** (scRMSD<2.0Å) and **Des.** (scTM>0.5). We also evaluate the diversity and novelty of generated protein structures. For **diversity**, we report the average pairwise TM-score of designable samples (scTM>0.5) among 50 generated samples. For measuring the novelty of a design, we identify the most similar known structure to the designed protein within the Protein Data Bank (PDB) and record its TM-score. We report an average of this value over designable samples among 50 generated samples as **Novelty**. For co-design methods, we directly predict the structure of the generated sequence using ESMFold (once) and compute all the above metrics based on the folded structure and generated structure. For sequence design methods, we do not report the above metrics since no structure is generated. Note that, for a method whose $Des$ is lower than 0.04, we do not its diversity and novelty. To evaluate the quality of the generated sequence, we report perplexity (**ppl**) from an autoregressive protein

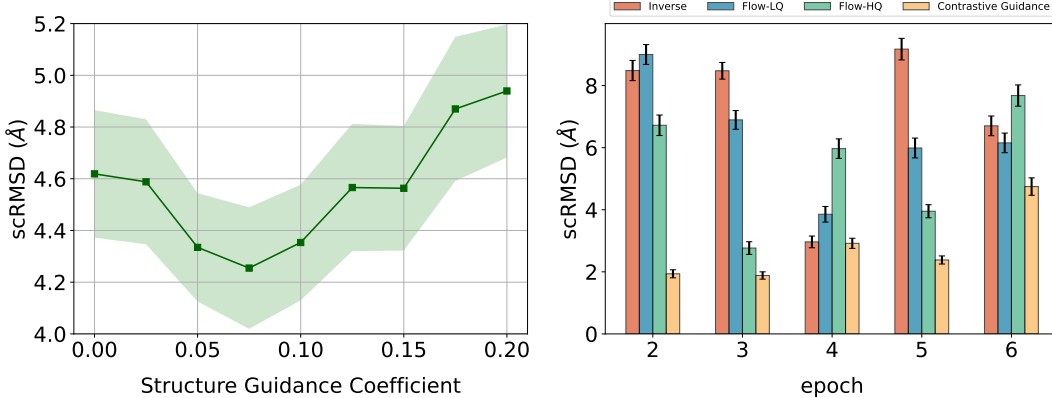

Figure 3: Left: scRMSD (Å) changes over the structure guidance coefficient. Right: scRMSD (Å) of our approach with dual flows fine-tuned for different epochs. "Inverse" stands for the guidance with the switched roles of FLOW-LQ and FLOW-HQ, i.e., using FLOW-HQ to calibrate FLOW-LQ.

language model (ProGen2 (Nijkamp et al., 2023)) to quantify if the patterns of generated sequences lie in natural sequence distribution and report ESMFold predicted **pLDDT** scores for structural plausibility. For backbone design methods, since no sequence is generated, we do not report these two metrics. See more details in App. A.

### 4.2 MAIN RESULTS

The evaluation metrics on length-400 proteins of all the baselines and our method are reported in Table. 1. Our method performs the best on scTM, scRMSD, Des.*, and Des. over all the methods. This demonstrates our hypothesis introduced in Sec. 3.2 and the effectiveness of contrastive guidance. Notably, our dual flow approach outperforms its counterpart (also its base model) by a significant margin in designability-related metrics. Our method shows similar diversity and novelty to its counterpart, Multiflow, though these two metrics of our method slightly fall behind the other methods. This also meets our expectation since classifier-free guidance for diffusion and flow models (Ho & Salimans, 2022; Zheng et al., 2023) also suffers from similar issues. Our methods also show competetive performance in evaluation metrics related to sequence quality, which indicates that our contrastive guidance in categorical distribution can effectively improve the plausibility of the generated sequence. The results on length-500 proteins are reported in Table. 2. Many methods totally fail under this setting. Our method still performs the best over all the methods, even though FLOW-HQ and FLOW-LQ are not fine-tuned on length-500 proteins as introduced in Sec. 4.1. This demonstrates the generalizability of our approach. See generated examples in App. B.

### 4.3 ABLATION STUDIES

We conduct ablation studies on the effects of the contrastive guidance coefficients $\lambda$ on the performance in terms of designability. In practice, for calibrating the probability flow on protein sequences, in addition to this coefficient, we also introduce sampling temperature, a widely-used coefficient in sequence generation to control the smoothness of a categorical distribution, to estimated posterior in Eq. (19). Results of the effects of different coefficients of sequences / structures contrastive guidance are shown in Fig. 2 and Fig. 3, respectively.

## 5 CONCLUSION

In this work, we introduced a contrastive guided sampling algorithm with dual multimodal flows to sample both sequences and structures of highly designable proteins, effectively addressing the challenges inherent in high-dimensional generative modeling of long proteins.

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

## A   METRICS

**TM score.** The template modeling score or TM score serves as a metric for assessing the resemblance between two protein structures. The TM score indicates the similarity between two structures by a score between $(0, 1]$, with 1 denoting a complete match (hence, higher score indicate greater similarity). TM score between two protein structures is defined by

$$\text{TM score} = \max \left[ \frac{1}{D_{\text{norm}}} \sum_{i}^{D_{\text{common}}} \frac{1}{1 + (\frac{d_i}{d_0(D_{\text{target}})^2})} \right], \tag{21}$$

where $D_{\text{target}}$ is the length of the amino acid sequence of the target protein, $D_{\text{query}}$ is the length of the amino acid sequence of the template protein, and $D_{\text{common}}$ is the number of residues that appear in both the template and target structures. $D_{\text{norm}} := D_{\text{common}}$ for "altmscore", $D_{\text{norm}} := D_{\text{target}}$ for "qtmscore", and $D_{\text{norm}} := D_{\text{template}}$ for "ttmscore". $d_i$ is the distance between the $i$th pair of residues in the template and target structures, and $d_0(D_{\text{target}}) = 1.24 \sqrt[3]{D_{\text{target}} - 15} - 1.8$ is a distance scale that normalizes distances.

**RMSD.** The root mean square deviation of atomic positions, commonly known as RMSD, quantifies the average distance between atoms (typically backbone atoms) of aligned molecules. When examining globular protein conformations, researchers typically assess three-dimensional structural similarity by calculating the RMSD of the $C_\alpha$ atomic coordinates following optimal rigid body superposition.

$$\text{RMSD} = \sqrt{\frac{1}{D} \sum_{i}^{D} \delta_i^2}, \tag{22}$$

where $\delta_i$ is the distance between atom $i$ and either a reference structure or the mean position of the N equivalent atoms. This is often calculated for the backbone heavy atoms $C$, $N$, $O$, and $C_\alpha$ or sometimes just the $C_\alpha$ atoms.

## B   VISUALIZATION OF GENERATED EXAMPLES

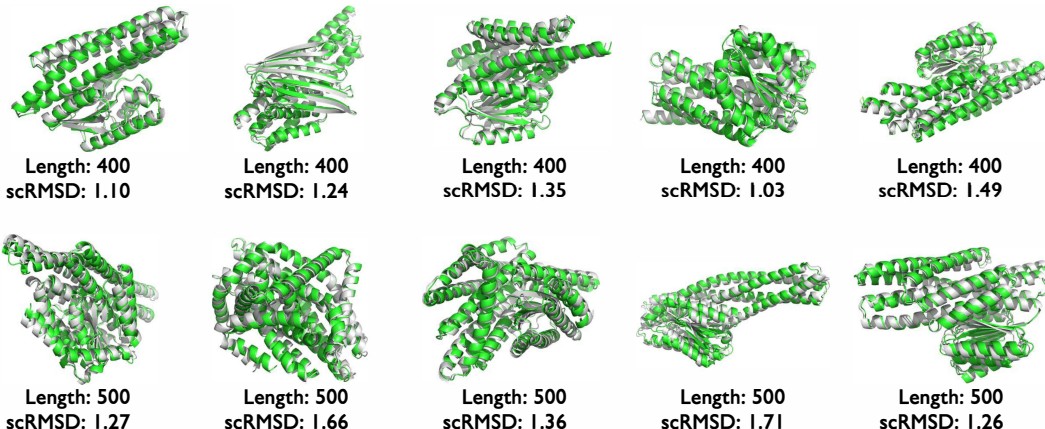

Figure 4: Examples of generated structures in green compared to refolded structures (generated sequence → ESMFold) in grey. Samples with scRMSD < 2 Å for lengths 400 (top) and 500 (bottom).

