# OpenReview forum: "Dual Flows with Contrastive Guidance for Generating Highly Designable Proteins"
_ICLR.cc/2025/Conference — Submitted to ICLR 2025_

### Official Review · Reviewer_JDZq · 2024-11-01

**Soundness:** 1
**Presentation:** 3
**Contribution:** 2
**Rating:** 3
**Confidence:** 4

**Summary:**

This paper introduces a method to guide flow-based protein generative model by contrasting a flow of models trained to low-quality and high-quality subset of self-synthesized dataset. The comparisons with baseline models and the inversely guided model shows that the proposed method can better generate longer proteins.

**Strengths:**

1. The framework is theoretically sound and is clearly stated for all modalities that the model utilizes.
2. Not much protein design papers tackles the sampling time guidance; this paper tackles one of the most important future directions of protein generative models. And given that calculating the designability of intermediate timestep samples is hard, I think it is effective and clever to contrast two models than employing some kind of classifier.

**Weaknesses:**

1. How useful is generating long monomer proteins?
2. Could you compare with Proteus [1]? It is published in same conference with MultiFlow; ICML 2024. From what I know, Proteus generates length-400 and 500 proteins much better than the proposed method. (~0.9 for length-400 and ~0.75 for length-500).
3. I am not persuaded that the performance increase entirely comes from contrastive guidance for two reasons:
    1. If you are using two models for your sampling, the fair comparison would be generating 2x more proteins and evaluate top half for baselines. How baselines perform in this setting?
    2. The tradeoff between designability and diversity is not clearly explained. For example, you can just boost up your designablity 2x by just generating similar proteins 2x.
        1. The proposed pairwise TM score doesn’t really give the sense of diversity. Can you explain the reasoning behind de-duplication and randomly selecting 50 proteins? It is really hard to interpret the numbers given all these procedures.
        2. I believe more common way of computing diversity is number of clusters. (as in MultiFlow [2], Proteus, Genie2 [3], and so on) Could you follow this evaluation scheme and provide the number of clusters?
4. Please explain the error bar and +- in the Figure and Tables.


[1] Wang, Chentong, et al. "Proteus: exploring protein structure generation for enhanced designability and efficiency." (2024)

[2] Campbell, Andrew, et al. "Generative Flows on Discrete State-Spaces: Enabling Multimodal Flows with Applications to Protein Co-Design." (2024)

[3] Lin, Yeqing, et al. "Out of Many, One: Designing and Scaffolding Proteins at the Scale of the Structural Universe with Genie 2." (2024)

**Questions:**

1. Why do we need Flow-HQ? Why train and use separate Flow-HQ model if you know self distillation training deteriorates the performance? Can you replace Flow-HQ to baseline MultiFlow (i.e. contrasting baseline MultiFlow and Flow-LQ)? How does it performs?
2. What setting did you use to plot Fig 3 (right)? What length and how many samples? Also can you add baseline MultiFlow performance as line?

---

### Official Review · Reviewer_bQrH · 2024-11-02

**Soundness:** 3
**Presentation:** 3
**Contribution:** 3
**Rating:** 6
**Confidence:** 4

**Summary:**

This paper incorporates the concept of contrastive guidance into the flow generation process. Specifically, it uses two flow models: one trained on high-quality data focusing on designable proteins (called the HQ model) and another trained on broader synthetic data, which may include non-designable examples (called the LQ model). During the generation process, the vector field is adjusted to steer away from the directions indicated by the LQ model and toward the HQ model. This guidance mechanism helps improve the designability of the generated proteins.

**Strengths:**

- The idea is good, incorporating the concept of contrastive guidance into the flow generation process. This approach of leveraging dual models (LQ and HQ) to steer the generation process has potential generalizability to other domains beyond protein design.

- The application domain, protein design, is highly impactful due to its significance in biotechnology and medical research. The experimental results are compelling, demonstrating significant improvements in designability metrics, such as scTM and scRMSD, compared to existing methods.

**Weaknesses:**

- The explanation regarding the statement “… avoid non-designable regions by gently steering it during sampling” could be clearer. It appears that the LQ model’s influence extends to all regions uniformly, rather than specifically targeting non-designable areas. It would be helpful if the authors could provide more detail on how the LQ model selectively identifies and influences non-designable regions during the sampling process.

 - The LQ model is fine-tuned on synthetic data, which likely includes a significant portion of designable proteins. This raises a concern that the vector field of the LQ model may not always exert a negative influence. Additionally, Figure 3 suggests that the LQ model’s impact may not always be detrimental, which could warrant further discussion.

**Questions:**

- Do you believe that the concept of contrastive guidance, as applied in this paper, could be generalized to other domains beyond protein design? If so, what potential challenges might arise in adapting it?

 - Why did you choose not to fine-tune a classification or regression model to predict the designability of proteins? Such a model could potentially filter out non-designable proteins from a large set of generated samples, enhancing the overall design process. Could you elaborate on this decision or provide insights into any limitations that might have influenced this approach?

---

### Official Review · Reviewer_TZav · 2024-11-02

**Soundness:** 2
**Presentation:** 3
**Contribution:** 2
**Rating:** 5
**Confidence:** 4

**Summary:**

This paper proposes a training and sampling algorithm called contrastive guidance which guides the sampling trajectory towards predictions of the "better" model and away from predictions of the "worse" model. They use this principle to finetune a pre-trained protein cogeneration model (Multiflow) and generate two checkpoints, one that is trained on synthetic protein structures of length 400 generated by the pre-trained model, and one that is only trained on the high-quality structures as judged by designability (scRMSD < 2A). They then sample protein sequences and structures with high designability from their checkpoint at lengths 400 and 500.

**Strengths:**

1. **Clarity**: The authors motivate their method clearly and it is easy to follow their argument as to why they arrived at the final method.
2. **Benchmarks**: The authors perform comprehensive benchmarks of baseline models for the different approaches (structure-based, sequence-based and cogeneration). They report these numbers for lengths 400 and 500 and give a comparison between different evaluation protocols (1 vs 8 ProteinMPNN sequences) as well as error bars for these numbers.
3. **Performance**: The authors show that their method on average has a lower average scRMSD and a higher designability score than the baseline model.

**Weaknesses:**

1. **Novelty**: The idea to combine predictions of a high-quality and a low-quality model to improve predictions has been proposed in [Karras et al. (2024)](https://arxiv.org/abs/2406.02507) under the name of **autoguidance**. Except for the minor difference that the authors here use a flow matching loss instead of a score matching loss , the main idea is very similar (although the application is new). A discussion of the relation of autoguidance to the proposed method would benefit the paper. Especially the theoretical and experimental analysis as to why it works could be helpful since this part is not covered deeply in this submission. One interesting aspect in Karras et al is for example the notion that CFG eliminates outliers and samples more from the base of the respective distributions, whereas autoguidance does not drop any significant part of the data distribution. Comparing these considerations with the discussion about the effect of contrastive guidance here could strengthen the theoretical motivation of this paper.
2. **Performance of baselines**: The performance of the baseline methods in Table 1 and Table 2 is worse than expected from other publications. This is unexpected, especially since recent benchmarking efforts like [ScaffoldLab](https://www.biorxiv.org/content/10.1101/2024.02.10.579743v3) and [ProteinBench](https://arxiv.org/abs/2409.06744) also show significantly better numbers for the baseline methods than suggested here in this paper. For example, in the setting of length 400/500 proteins and 8 ProteinMPNN samples, ProteinBench (whose benchmark is publicly reproducible) reports scRMSD scores of 2.12/xx for FrameFlow while this paper here reports 6.12. It reports them with an error margin of 4.94, which in theory includes that other measurement. The question is how strong the claim of improved performance can be made in that setting then, especially given that the pre-finetuned baseline Multiflow has a score of 2.71 with an error of 3.65. The authors are encouraged to check their numbers against these publicly available benchmarks and comment on the discrepancy and the validity of their conclusions based on these high error margins. Possible action items to strengthen the claims of the authors here could be
    - Provide a detailed comparison table showing their results side-by-side with the ProteinBench and ScaffoldLab results for the same methods and metrics.
     - Directly compare their evaluation protocol to that used in ProteinBench and ScaffoldLab, highlighting any differences.
     - If possible, re-run evaluations using the exact protocols from these public benchmarks.
     - Discuss how the large error margins impact the statistical significance of the performance claims.
3. **Selection of evaluation lengths**: The authors only show evaluations for lengths 400 and 500, not for shorter lengths. Since most of the baseline methods were only trained on proteins of length up to 256 or sometimes 384 residues, but the authors train their finetuning model exclusively on samples of length 400, it is not clear whether the claimed improvement comes from their contrastive guidance framework or just from the additional training on long sequences and structures. Additional results for shorter lengths would strengthen the claims made by the authors here.
4. **Performance tradeoffs**: The authors show that their model outperforms the previous state-of-the-art methods on designability (however with a high error bar), but also show that they are worse on diversity than other SOTA methods and even worse than their baseline model Multiflow. Since the designability-diversity trade-off of these methods can be tuned with factors like sampling temperature, it does not seem clear to me that this method actually improves the overall "Pareto frontier" of this designability-diversity trade-off. The authors are encouraged to present designability and diversity numbers at different temperatures for the different methods to strengthen their SOTA claims. If possible, include experiments or analyses that attempt to match the diversity of baseline methods while comparing designability and the other way around.

**Questions:**

1. Quote: "residue frame representation has demonstrated the best performance" (page 1). Is their any evidence that residue frame representation is better than other representations in some benchmark, besides the fact that some SOTA models happen to use that representation?
2. In Figure 3b, it seems like the efficacy of contrastive guidance gets worse over training time. Do you have any intuition why this is the case?
3. In Figure 2, it is visible that the best scRMSD values are achieved with a low sampling temperature and a hihg sequence guidance coefficient. During sampling for the benchmarks in Table 2 and 3, do you use the same sampling temperatures for Multiflow as for your method? It would be interesting to see the numbers for both methods under a range of sampling temperatures to clarify if contrastive guidance advanced the Pareto frontier" of the overall designability-diversity tradeoff.

---

### Official Review · Reviewer_AZDs · 2024-11-04

**Soundness:** 3
**Presentation:** 2
**Contribution:** 2
**Rating:** 5
**Confidence:** 3

**Summary:**

The authors demonstrate that by carefully curating high- and low-quality generated samples from Multiflow, they can fine-tune the model into separate high-quality and low-quality versions. They then use contrastive guidance to steer the sequence/structure co-generation (in R3, SO3, and discrete spaces) towards producing high-quality samples. Empirical results show that this approach  improves designability over standard Multiflow, especially for long proteins in the co-design task.

**Strengths:**

The contrastive guidance approach is well-motivated, conceptually sound, and, to my knowledge, novel—especially as applied to protein co-generation.
Sequence/structure co-generation is an important, largely unresolved problem. The results presented here indicate some viability for this approach (with certain limitations) particularly for larger proteins.
The paper is generally well-written, and the results are adequately presented.

**Weaknesses:**

1. One key issue with Multiflow is that its sequence/structure co-design does not outperform a simpler approach: designing the backbone and then performing inverse folding. For any substantial improvement, it must be demonstrated that co-design offers advantages over this backbone-first approach (+ inverse folding). Currently, the manuscript lacks a comparison between co-design and PMPNN, which is essential. Additionally, a major advantage of Multiflow is its capacity to handle multiple modalities, such as inverse folding and forward folding (though performance in forward folding is limited). The manuscript does not adequately discuss these other modalities. I would consider raising the score if these additional results were presented.
2. Designability is a problematic metric to optimize (e.g., ESMFold has numerous false negatives, and its failure to refold a structure does not necessarily imply that the structure is incorrect). This reliance on the Flow-HQ dataset, which only includes structures foldable by ESMFold, may therefore be flawed. Designability also has a bias toward alpha helices (partially because folding algorithms handle them better). Once designability reaches a certain level, other metrics become more important. Predictably, the current method suffers from reduced diversity and novelty, which should be discussed further. Additionally, the metric of novelty in this paper is somewhat inadequate. It should measure similarity not only to PDB data but also to synthetic datasets, examining how closely generated samples resemble the training distribution. A secondary structure analysis should also be included, especially as Figure 4 primarily shows alpha helices.
3. Note that Multiflow already performed distillation, which substantially improved designability. I anticipate further improvements could be achieved with careful tuning of distillation procedures. Consequently, I do not find the empirical results particularly compelling; for instance, in Figure 3 (right), at epoch 3, simple fine-tuning on designable samples reaches almost the same performance as contrastive guidance.

**Questions:**

1. I am surprised that fine-tuning on only designable samples does not consistently improve scRMSD (Figure 3, right). scRMSD appears unstable across epochs—can you explain this variability?
2. The results focus on proteins with lengths of 400 and 500, but proteins of these lengths are rarely synthesized for practical reasons. How does the model perform with shorter proteins (50–300)?

---

### Meta-Review · Area_Chair_n6hJ · 2024-12-20

**Metareview:**

Three out of four reviewers recommend to reject this paper, and one reviewer judged this paper to be marginally above the acceptance threshold. The authors have not provided a rebuttal. Therefore, I recommend to reject this paper.

**Additional Comments On Reviewer Discussion:**

There was no reviewer discussion given that no rebuttal was provided.

---

### Decision · Program_Chairs · 2025-01-22

Reject